# Species Identification of Wireworms (*Agriotes* spp.; Coleoptera: Elateridae) of Agricultural Importance in Europe: A New “Horizontal Identification Table”

**DOI:** 10.3390/insects12060534

**Published:** 2021-06-08

**Authors:** Lorenzo Furlan, Isadora Benvegnù, María Fabiana Bilò, Jörn Lehmhus, Enrico Ruzzier

**Affiliations:** 1Veneto Agricoltura, Viale dell’Università 14, Legnaro, 35020 Padova, Italy; fabiana.bilo@venetoagricoltura.org; 2Freelance, Via. G. Mameli 13, 45011 Adria, Italy; isadora.benvegnu@gmail.com; 3Institute for Plant Protection in Field Crops and Grassland, Julius Kühn-Institute (JKI)-Federal Research Centre for Cultivated Plants, Messeweg 11-12, 38104 Braunschweig, Germany; joern.lehmhus@julius-kuehn.de; 4Department of Agronomy, Food, Natural Resources, Animals and the Environment (DAFNAE), University of Padova, Viale dell’Universita 16, Legnaro, 35020 Padova, Italy; enrico.ruzzier@unipd.it

**Keywords:** beetles, click beetles, crops, IPM, larvae, pest

## Abstract

**Simple Summary:**

Wireworms are soil-dwelling larvae that damage multiple arable crops. The most common wireworms found in European cultivated fields belong to the genus *Agriotes*. Large amounts of insecticides are applied on a prophylactic basis to control them; before any treatment can be applied, however, legislation imposes an assessment of whether pest population levels exceed a damage threshold above which a significant yield reduction is expected. Thresholds vary greatly among species, thus quick and reliable larval identification is needed. This will result in the implementation of the appropriate integrated pest management practices. Furthermore, research into non-chemical strategies involves carrying out tests with live wireworms identified to species. Wireworms were observed to identify live larvae rapidly on site to assess density and compared with species-specific thresholds before sowing, and for laboratory experiments to be performed. This work led to a “synoptic key” that reliably identifies most live larvae, while traditional keys consider only single characters step by step. The key considers several discriminating morphological characters in order of stability. Identification becomes reliable when at least two main discriminating characters are found and attributed to a single species.

**Abstract:**

Wireworms are yellowish soil-dwelling larvae that damage a wide range of arable crops. The most common wireworms found in European cultivated fields (except for the Caucasus) belong to the genus *Agriotes* (Coleoptera: Elateridae). In several European countries, environment-impacting insecticides are applied on a prophylactic basis to control them. However, before any treatment can be applied, European legislation requires that an assessment is done when pest population levels exceed a damage threshold. The threshold substantially depends on wireworm species, thus quick reliable larval identification is needed to implement the appropriate integrated pest management practices. Furthermore, research into non-chemical strategies involves carrying out tests with live and identified wireworms. Thus, thousands of wireworms were observed in a bid to identify live larvae so that larval density could be assessed and compared with species-specific thresholds before sowing, and laboratory experiments were carried out. This work led to a horizontal identification table that allows for quick and accurate identification of live larvae. This key, unlike traditional dichotomous keys, simultaneously considers a set of multiple discriminating morphological characters in order of stability. The key can be reliably used by less experienced users and, once minimum familiarity is acquired, most larvae can be identified rapidly, with high precision.

## 1. Introduction

Wireworms, the larvae of click beetles (Coleoptera: Elateridae), rank among the major soil pests of a large number of arable crops in Europe and North America [1]. The most harmful species in Europe are in the genus *Agriotes* Eschscholtz, 1829: *Agriotes brevis* Candèze, *A. lineatus* L., *A. litigiosus* Rossi, *A. obscurus* L., *A. proximus* L., *A. rufipalpis* Brullé, *A. sordidus* Illiger, *A. sputator* L. and *A. ustulatus* Schäller (Elateridae: Elaterinae: Agriotini Champion, 1894) [2]. *Agriotes lineatus*, *A. obscurus* and *A. sputator* were first described as a complex of species [3] and have been confirmed as major soil pests multiple times [4]. *Agriotes brevis*, *A. litigiosus*, *A. sordidus* and *A. ustulatus* are the major crop-damaging species in Italy’s Po Valley [5,6]. *Agriotes sordidus* became a major pest in western Germany [4] after being reported in France [7]. This species was found to be the most frequent cause of severe damage to maize crops, alongside *A. brevis*, in long-term research in northeast Italy [8]. *Agriotes brevis* is a major pest in Italy, as reported above, as well as in Eastern European countries [9,10,11,12,13]; it is also present in Western Europe [14]. *Agriotes ustulatus* is a major pest in Central and Eastern Europe [4,9,10,12,15] while *A. litigiosus* is a major pest in Italy, Greece and several Eastern European countries [9,10,16,17,18].

Current European legislation, including Directive 128/2009/EC, makes it compulsory to use integrated pest management (IPM) practices, which require farmers to assess whether pest population levels exceed the damage threshold before applying any treatment. Wireworm damage risk [8] and thresholds [19] can change depending on *Agriotes* species. The efficacy of biological control with entomopathogens is also species dependent. Therefore, it is important to correctly identify the species causing the damage and do so rapidly so that IPM may be implemented immediately. Another important IPM principle establishes that when damage thresholds are exceeded, non-chemical pest control methods should be considered as a replacement for chemical treatment. Setting up non-chemical methods to control wireworms, such as agronomic strategies [20], biocidal plant materials [21] and entomopathogens [22], involves carrying out tests in laboratories and/or in semi-natural conditions with larvae which need to be kept alive during identification [20].

*Agriotes* larvae can be accurately identified using PCR and DNA barcoding [23,24,25,26], however, these techniques are not immediate and often imply the destruction of the specimen.

In the past, some taxonomic keys were designed for Elateridae larvae. The most complex and complete were those devised by Dolin [27,28], although others including *Agriotes* have also been published [29,30,31,32]. All of the aforementioned keys combine a number of characters in a rigid dichotomous method that does not consider the huge variability in discriminating characters, as indicated by Eidt [33], Lehmhus and Niepold [26], Oehlke [34] and by the over 35 years of observations performed by the first author. Dichotomous keys can only, with some difficulty, take into account the variability of certain traits and therefore lose relevant information.

Furthermore, currently available dichotomous keys include species of non-agricultural habitats but may lack some of the agriculturally relevant taxa (e.g., Klausnitzer [30]; Rudolph [32]). Therefore, this manuscript aims to bridge the existing identification gap through a horizontal identification table which allows the identification of live *Agriotes* larvae effectively and rapidly, and also encompasses intraspecific variability. Horizontal tables have only occasionally been used before in entomology, e.g., Stehr [35]. In particular, the horizontal table provided here is intended to help the identification of *Agriotes* larvae affecting a wide range of field and vegetable crops in Europe (with the exception of the Caucasus). The table presented here appears to be the best available solution for a morphological key to a restricted number of species which are quite variable in several larval traits.

## 2. Materials and Methods

This section is subdivided into two different steps: the first preparatory step consisted of long-term observations to collect a large series of morphological data on *Agriotes* larvae; the second step involved evaluating data and creating the “horizontal identification table”.

### 2.1. First Step: Larvae Description per Species

Research was carried out from 1985 to 2020 based in Italy and in parallel from 2009–2020 based in Germany. The first few years were concentrated on directly observing larvae and creating the first morphological dataset in order to separate species. Data recording was performed with three major actions:Elateridae larvae collected in the field in several European countries (listed below) were reared until the adult stage in single vials, according to the method described by Furlan [36,37]. The rearing was performed in order to associate the larva (and its morphological features) with the adult. Images were identified by Giuseppe Platia, a leading expert in Palearctic Elateridae [38];rearing cages were built and used to house identified beetles for egg deposition [36];click beetles were collected in the field in Germany, identified by keys [39,40] and wireworms reared from these beetles following the method by Kölliker et al. [41]. The larval specimens from known species were then checked for consistency with existing keys [7,27,30,31] and also compared with field-collected larvae, mainly from Germany.

In the subsequent years, over 50,000 larvae from several European countries were identified and described based on an initial description of the larvae and an initial provisional discriminating key designed around a number of potentially reliable characters. At the same time, larvae obtained from eggs were characterized in order to evaluate the variability of their main discriminating characters.

At the end of the research, L.F. observed more than 10,000 larvae of each of the following species: *A. brevis* (95% from northeast Italy, the rest from central Italy, France and Eastern European countries); *A. litigiosus* (90% from northeast Italy, 5% from central and southern Italy, the rest from Greece); *A. sordidus* (90% from northeast Italy, 5% from central and southern Italy, the rest from France and Germany); *A. ustulatus* (96% from northeast Italy, the rest from Eastern European countries and Germany); more than 300 larvae of *A. obscurus* and *A. lineatus* (50% from Italy: 20% from central and southern Italy, the rest from Central and Eastern European countries); and 5 to 20 specimens for the remaining species. Additionally, between 2009 and 2020, J.L. identified about 3000 *A. lineatus* wireworms, 5000 *A. obscurus,* 4000 *A. sputator*, 900 *A. ustulatus* and about 300 *A. sordidus*, mainly from Germany. Specimens from Romania were also identified (*A. ustulatus* plus a few *A. rufipalpis*). Larvae were observed and measured under a binocular microscope with a micrometer or a Keyence photo microscope with micrometer and angle-measuring functions; observation magnification ranged from 20× to 200× according to the character under observation.

Voucher specimens studied in this phase of the research were deposited in the collection of Lorenzo Furlan (via Quintino Sella 12, 30027 San Donà di Piave VE, Italy) and Jörn Lehmhus (Institute for Plant Protection in Field Crops and Grassland, Julius Kühn-Institute (JKI)—Federal Research Centre for Cultivated Plants, Messeweg 11–12, 38104 Braunschweig, Germany).

Drawings were realized by Fabiana Bilò; photos were taken by Fabiana Bilò and Jörn Lehmhus.

### 2.2. Second Step: Discriminating Characters

*Agriotes* larvae can be easily distinguished from all other European elaterid genera due to the two spiracles on the dorsal part of their 9th abdominal segment [29].

The body parts chosen to distinguish larvae of the different *Agriotes* species were:(1)the frontoclypeus (clypeus), in relief on the head capsule (Figure 1 and Figure 2a);(2)the mandibles (Figure 2a);(3)the abdominal segments (dorsal and side part including spiracles; Figure 3a);(4)the 9th abdominal segment (Figure 3b).

For each of the abovementioned body parts, the following characters were considered:*Clypeus*: absence/presence and density of the punctures on the clypeus; amplitude of the measured angle (ma) (Figure 2a). Some taxa present a certain level of intraspecific variability in the amplitude of the angle, with a certain overlap in amplitude being found in a few species.*Mandibles*: absence/presence of the subapical tooth on the inner margin of the mandible. Amplitude of the measured angle (ma) between the subapical tooth (sbt) and the apex of the mandible (st) (Figure 2b). This character is often difficult to observe because the tooth tends to erode due to larval feeding; consequently, only recently formed mandibles on post-molting larvae, or larvae with slightly worn mandibles, can be reliably identified.*Abdominal segments*: absence/presence of granules and punctures on the tergites; length/width ratio of spiracles; presence/absence of *minus setae* (ms) (Figure 3a).*Ninth abdominal segment*: general shape; absence/presence of the bulge at the apex of the segment; profile and proportion of the bulge when present; absence/presence of dark elevated setiferous tubercles; presence and shape of terminal spike (ts) (Figure 3b).

Each of the above characters was carefully observed in the specimens associated with an identified adult and defined after comparison with drawings/descriptions by previous authors [7,26,27,28,29,31,32,33] when available.

The main characters separating the species were established and their stability was assessed by looking at all the available specimens collected over the years.

The characters were then divided into two groups:**Primary characters**: high stability (up to 100% of the observed specimens): present in just one or a few species only.**Secondary characters**: lower stability and specificity, but useful for confirming identification when combined with primary and other secondary characters.

## 3. Results

### 3.1. Species Characters

The diagnostic characters of the species are reported in descending order, starting from primary ones, and then moving onto secondary ones. The order in which the various parts of the body are presented for each species reflects the importance they have in supporting identification.

#### 3.1.1. *Agriotes ustulatus* (Schäller 1783)


**Primary characters**


*Ninth abdominal segment*. Setiferous tubercles present and well demarked (feature unique to this species) (Figure 4a,b): by looking from the apex of the dorsal part of the 9th segment and moving towards its base, each seta arises from an elevated dark tubercle. From a lateral view, it appears as a dark elevation around the base of the seta, while from above it resembles a circle.


**Secondary characters**


*Ninth abdominal segment*. Segment subconical in both dorsal and lateral view, suddenly shrinking in the apical third (Figure 4a,b).*Clypeus*. *Agriotes ustulatus* is the only species in which the measured angle is consistently >90° (Figure 4c). This angle was >90° on both sides of the clypeus in about 99.5% of the specimens observed (100% when we consider only one of the two sides), while it was equal to 90° in the remaining 0.5%. The same character was found in *A. sputator* although with a lower consistency (75% of specimens).*Mandibles*. Subapical tooth present, forming a measured angle clearly >90° (Figure 4e,f) with the apex of the mandible; about 15% of the specimens possess an angle of approximately 90°. Depending on the larva’s origin area, the angle varies between approximately 100° and 150°, with an average of 120°. The measured angle is generally more obtuse in *A. ustulatus* than in *A. brevis*.

#### 3.1.2. *Agriotes sordidus* (Illiger 1807)


**Primary characters**


*Ninth abdominal segment*. Segment subconical in both dorsal and lateral view, suddenly shrinking in the apical fourth part (Figure 5a); terminal spike of the segment presenting a large square protrusion (bulge) at the base (Figure 5b). Proportions and size of the bulge vary among individuals, but it is always present and easy to spot. The peculiar bulge illustrated identifies *A. sordidus* with certainty in the regions of its distribution.*Mandibles*. The subapical tooth forms a measured angle far greater than 90° (Figure 5c,d); this angle differs from all the other species with the only exceptions being *A. rufipalpis* and some *A. ustulatus* (average angle of *A. sordidus*: 148°; range: 130–170°).


**Secondary characters**


*Clypeus*. Measured angles of the frontoclypeus form a right angle (Figure 5e,f); this feature is fairly consistent among individuals.*Other secondary characters*. Rarely, there can be granules on thoracic sternites between coxae, but smaller and weaker than in *A. sputator*.

#### 3.1.3. *Agriotes litigiosus* (Rossi 1792)


**Primary characters**


*Mandibles*. Almost falciform, without the subapical tooth on the inner margin. The character is 100% consistent and can be observed even in significantly worn specimens (Figure 6a,b).*Ninth abdominal segment*. Segment oblong, 2× longer than wide at the base. Apical part of the segment pointed (spike) and symmetrical; some specimens present a small asymmetrical caudal enlargement (bulge) prior to the terminal spike (Figure 6c–f). Character 100% consistent.


**Secondary characters**


*Clypeus*. Measured angles of the frontoclypeus of 90° or <90°. This character is fairly consistent among the specimens observed (Figure 6g,h).

#### 3.1.4. *Agriotes sputator* (L. 1758)


**Primary characters**


*First to eighth abdominal segment*. Antero-lateral part of the abdominal tergites more finely and densely punctured (almost grainy) than the remaining part of the tergite (Figure 7a,b). Character consistent (100% in our study, 98% in Lehmhus and Niepold [26]), exclusive to this species.*Thoracic sterna*. Thoracic sternites between the coxae present distinct granules (Figure 7c). The character is stable, but its observation is time-consuming and often complex. Generally, this character is most clearly visible on the last leg-bearing segment (3rd). Note: in one local population in northwest Germany, this character was missing in multiple specimens, which were identified as *A. sputator* by their other morphological traits and PCR results.


**Secondary characters**


*Clypeus*. Measured angles slightly >90°, on at least one side (>75% of the cases observed) (Figure 8c,d).*Mandibles*. Subapical tooth present (Figure 8d,e); measured angle ranging from 90° to 100° in Italy (in Germany, average angle: 95.6°; range: 85–112°).*Ninth abdominal segment*. Segment longer than wide at the base. Tergite densely and roughly punctured (Figure 8a,b), bearing longitudinal lines. Inner longitudinal lines (iim) were the same length as the outers (oim) (Figure 8a,b). Terminal spike of the segment longer than wide.*Size*. A fully grown larva of this species is substantially smaller than other species, with its head capsule usually <1.5 mm wide.

#### 3.1.5. *Agriotes brevis* (Candèze 1863)


**Primary characters**


*Ninth abdominal segment*. Inner longitudinal lines (iim) longer than outers (oim) and 0.5× the length of the whole segment (Figure 9a,b). The character was 100% consistent for at least one of the two inner longitudinal lines; it applied to both the inner lines in more than 80% of specimens.


**Secondary characters**


*Ninth abdominal segment*. Terminal spike of the segment longer than wide (Figure 9a,b).*Clypeus*. Measured angle variable: 90° in 17% of specimens, most commonly <90° (80% of specimens) or slightly <90° (3%) (Figure 9c,d).*Mandibles*. The subapical tooth is present, forming a measured angle slightly more than 90° (Figure 9e,f).*Size*. Mature larvae of this species are similar to those of *A. sputator*; however, *A. brevis* never exceed 20 mm in length and the width of the head capsule is <1.5 mm.

#### 3.1.6. *Agriotes obscurus* (L. 1758)


**Primary characters**


*Agriotes obscurus* (as *A. lineatus*) does not present any of the primary discriminating characters described for the previous species. Unlike *A. lineatus* (Figure 10e,f), its abdominal spiracles are clearly longer than wide, being narrowed posteriorly and slightly oblique (Figure 10c,d). This character is fairly consistent, although in some specimens it has to be carefully observed at high magnification (>50×). *A. lineatus* and *A. obscurus* are the two most difficult species to differentiate, and it may be impossible to identify single abraded specimens.


**Secondary characters**


*Ninth abdominal segment*. Inner longitudinal lines (iim) as long as outers (oim) and shorter than half of the segment length (Figure 11a,b). Segment 1.5–1.8× longer than wide at the base. Terminal spike of the segment as long as it is wide.*Clypeus*. Measured angles of approximately 90° (Figure 11c,d). Integument strongly punctured.*Mandibles*. Subapical tooth is present (Figure 10a,b), measured angle from 90–100° in Italy (in Germany, average angle: 101.3°; range: 86–128°).*Other secondary characters*. Spiracles on thorax and abdominal segments (1–8) 1.7× longer than wide (range: 1.3–2.1 in Germany in recent measurements). *Minus setae* present (Figure 10c,d). Rarely, there can be granules on thoracic sternites between coxae, but smaller and weaker than in *A. sputator*.*Body color.* On average, darker than *A**. lineatus*, but this character can be difficult or even impossible to evaluate in a single individual.

#### 3.1.7. *Agriotes lineatus* (L. 1767)


**Primary characters**


Like *A. obscurus*, *Agriotes lineatus* does not present any of the primary discriminating characters described for the other species. Unlike *A. obscurus* (Figure 10c,d), however, the abdominal spiracles of *A. lineatus* are shorter and more parallel-sided and subquadrate (Figure 10e).


**Secondary characters**


*Ninth abdominal segment*. Inner longitudinal lines (iim) as long as the outers (oim) and shorter than half the segment length (Figure 12a,b). Terminal spike of the segment as long as it is wide.*First to eighth abdominal segment*. *Minus setae* is usually absent on the abdominal segments (1–8) (Figure 10e,f). This character had a 97% consistency in our records (all or most of the abdominal segments did not have a *minus setae*), slightly higher than that reported by Lehmhus and Niepold [26] (93% average). However, this character can vary among populations and a high percentage of local specimens may occasionally possess the *minus setae*.*Clypeus*. Measured angles of about 90°; integument less punctured than *A. obscurus* (Figure 12c,d).*Mandibles*. Subapical tooth is present; measured angle usually ≤90° (Figure 12e,f); variable character due to abrasion of subapical tooth and apex of mandible.*Other secondary characters*. Spiracles on thorax and abdominal segments (1–8) on average 1.4× longer than wide.*Body color*. Lighter yellow than in the other species, but beware of freshly molted specimens from other species (lighter in color).

Identification can be considered highly reliable (>99%) when a specimen does not present any of the primary characters reported for the other species, but clearly shows two or more secondary characters. The measured angle of mandibles and/or the absence of the *minus setae* on the abdominal segments are especially important for identification. Reliability increases with the number of secondary characters observed.

#### 3.1.8. *Agriotes proximus* (Schwarz 1891) and *Agriotes rufipalpis* (Brullé 1832)

*Agriotes proximus* can be reliably separated from all the other *Agriotes* species but it is indistinguishable from *A. lineatus* (Figure 12), raising questions about its species status (as suggested by Staudacher et al. [25]). Tóth et al. [42] and Vuts et al. [43] observed that both pheromone profile and nucleotide sequence analysis suggest a close relationship between the two species, postulating a taxonomic revision. *Agriotes rufipalpis* (Figure 13) adults are clearly genetically [25] and morphologically distinguishable from *A. sordidus*, although both species are attracted by the same pheromone [44], but we could not find any character differentiating *Agriotes rufipalpis* larvae from those of *A. sordidus*. The two species, however, appear to be well-separated geographically, with *A. rufipalpis* being widespread throughout southeastern Europe (i.e., Bulgaria [45], Croatia [9,10], Greece [16], Hungary [18,22], Romania [9,10] with the northwesternmost outposts in the Slovak Republic [40]) and *A. sordidus* occurring in southwestern Europe (i.e., Italy [5,6,37,46], France [7] and Germany (easternmost finds in Germany in the region of Hildburghausen, south of the Thuringian Forest, Thuringia) [4]). The distribution overlap between *A. rufipalpis* and *A. sordidus* provided by Dolin [28] should be considered doubtful and requires further investigation.

### 3.2. Horizontal Identification Table

The diagnostic characters of the species are reported left to right in descending order, starting from the primary ones and moving onto the secondary ones. The order in which the various parts of the body are presented for each species reflects the importance they have in supporting identification (Appendix A).

### 3.3. Horizontal Identification Table—Usage

Discriminating characters and their stability are represented in a double entry identification table (Figure 14). This key should be read progressively, column by column, from left to right. Primary characters, grouped in the left-hand columns of the table, should be checked first. When the first character has been recognized, the other columns should be checked progressively along the line of the identified character. The contemporary presence of a set of characters in a single specimen makes identification highly reliable.

## 4. Discussion

This table allows the identification of European *Agriotes* crop pests with high reliability. There are primary characters for some species which will practically always result in correct identification. Furthermore, some secondary traits, often in combination with each other, can be very useful. However, when affected by abrasion, secondary traits can be inconclusive. This is the case for the measured angle when specimens have a subapical tooth on their mandibles and that can be reduced or even disappear due to abrasion [26,33,47,48]. In addition, several species have a certain percentage of “aberrant” individuals, which have one trait that normally occurs in other species. For example, Eidt [33] used the *minus setae* to distinguish between *A*. *lineatus* and *A*. *obscurus*, as it initially appeared to be a straightforward identification trait. However, occasionally, *Agriotes lineatus* has the *minus setae*, and *A. obscurus* is missing it [26]. Van Emden [29] considered the size and shape of the spiracles, but Oehlke [34] noted that the spiracle length and width varied, as did their quotient, leading to considerable overlap (1.3–2.2:1 in *A. obscurus*, including some *A. lineatus*, compared to 1.7–3:1 in *A. sputator* according to Oehlke [34]). These measurements are similar to ours: 1.1–1.7:1 in *A*. *lineatus*; 1.3–2.1:1 in *A*. *obscurus*; and 1.4–2.8:1 in *A*. *sputator*. A dichotomous key could be misleading in such cases of overlap between species, but a key taking into account a range of other traits can still identify a species with a contemporary set of multiple characters.

Oehlke [34] already attempted a parallel comparison of five different traits to distinguish between *A. lineatus*, *A. obscurus* and *A. sputator*. In contrast, our paper evaluated 17 different characters to distinguish between all harmful *Agriotes* species. This methodology enables each discriminating character to be compared almost immediately as it prioritizes some characters over others, with stable characters being considered first.

Our horizontal identification table allows quick and reliable identification in most cases. However, when only worn specimens of *A. lineatus* and/or *A. obscurus* are available, greater effort is required as mistakes may be made. Therefore, this key is an important additional tool for applied entomologists in need of a fast decision-making tool for IPM strategies, especially in cases of infestation.

Being able to handle and identify live larvae, which can be used for further experiments, is the other useful achievement provided by this key. The methodology used in this identification table may be progressively improved by researchers and applied technicians, with direct and indirect benefits for research, agriculture and environmental conservation.

## Figures and Tables

**Figure 1 insects-12-00534-f001:**
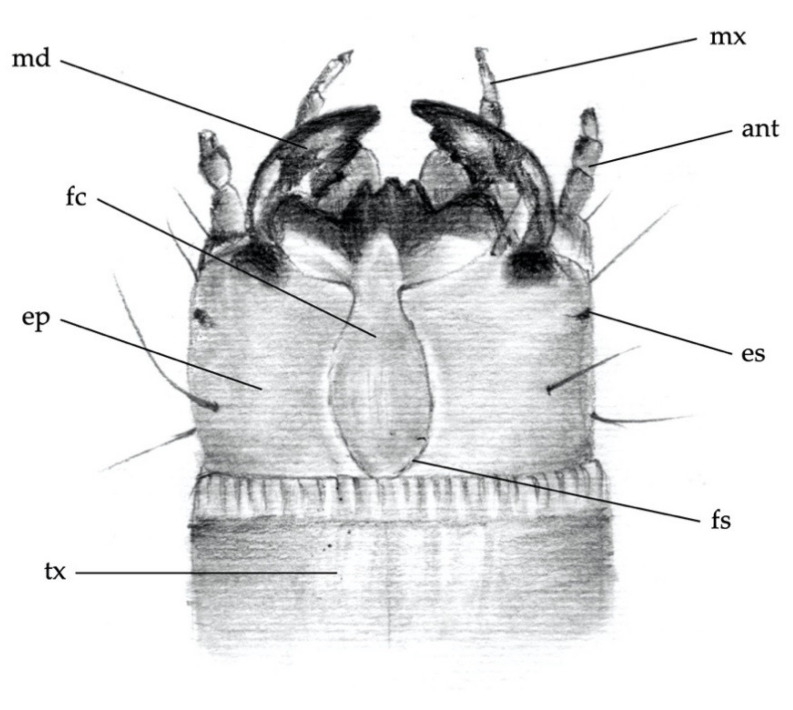
Anatomy of the larva. Dorsal head capsule and 1st thoracic segment: ant, antenna; ep, epicraneal plate; es, stemma; fc, frontoclypeus; fs, frontal suture; md, mandible; mx, maxillary palp; tx, prothorax.

**Figure 2 insects-12-00534-f002:**
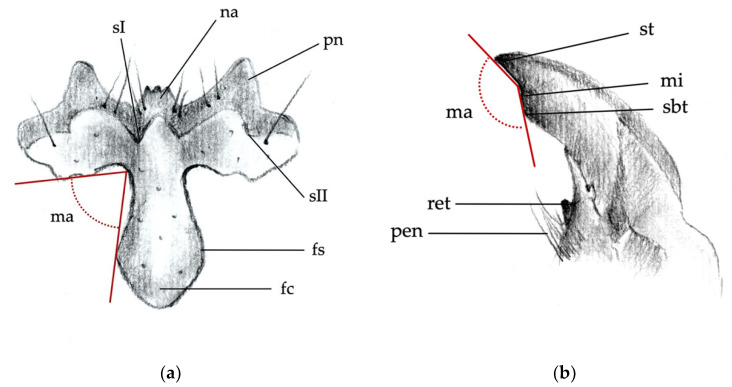
Main body parts distinguishing larvae of the different species. (**a**) Clypeus: fc, frontoclypeus; fs, frontal suture; ma, measured angle; na, nasal; sI, sII, sulci of the frontoclypeus; pn, paranasal lobe. (**b**) Right mandible: ma, measured angle; mi, inner margin of the apex; pen, penicillus; ret, retinaculum; st, apical tooth (apex of the mandible); sbt, subapical tooth.

**Figure 3 insects-12-00534-f003:**
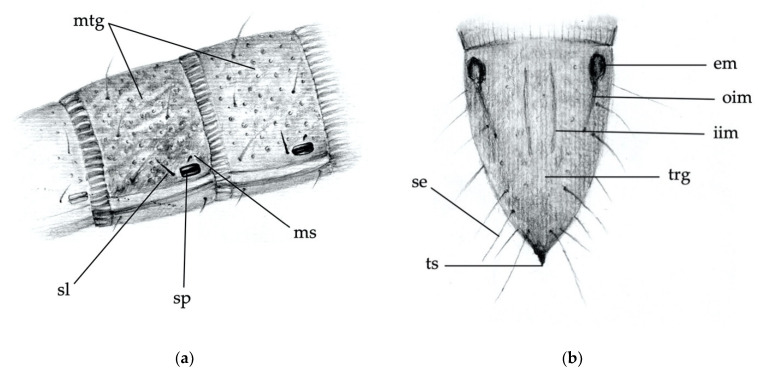
Main body parts distinguishing larvae of the different species. (**a**) Abdominal segments (dorso-lateral view): mtg, mediotergite with granulation and roughness; ms, *minus seta*; sl, large seta; sp, spiracle. (**b**) Ninth abdominal segment (dorsal view): em, “eye-like” impression; iim, inner longitudinal line; oim, outer longitudinal line; se, seta; trg, tergum; ts, terminal spike.

**Figure 4 insects-12-00534-f004:**
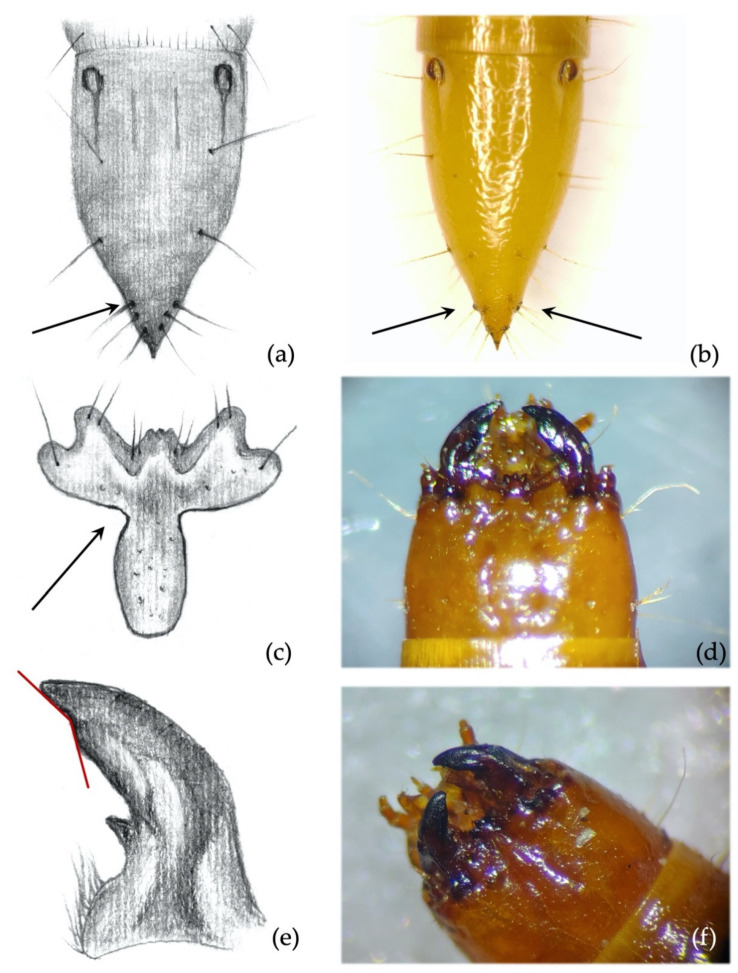
*Agriotes ustulatus* (Schäller 1783). Discriminating characters: (**a**,**b**) 9th abdominal segment with dark elevated tubercles around setae; (**c**) details of the frontoclypeus forming an angle >90° in about 99.5% of the specimens; (**d**) view of the head with frontoclypeus; (**e**) details of the right mandible with subapical tooth forming an angle clearly >90°; (**f**) view of the head with mandibles.

**Figure 5 insects-12-00534-f005:**
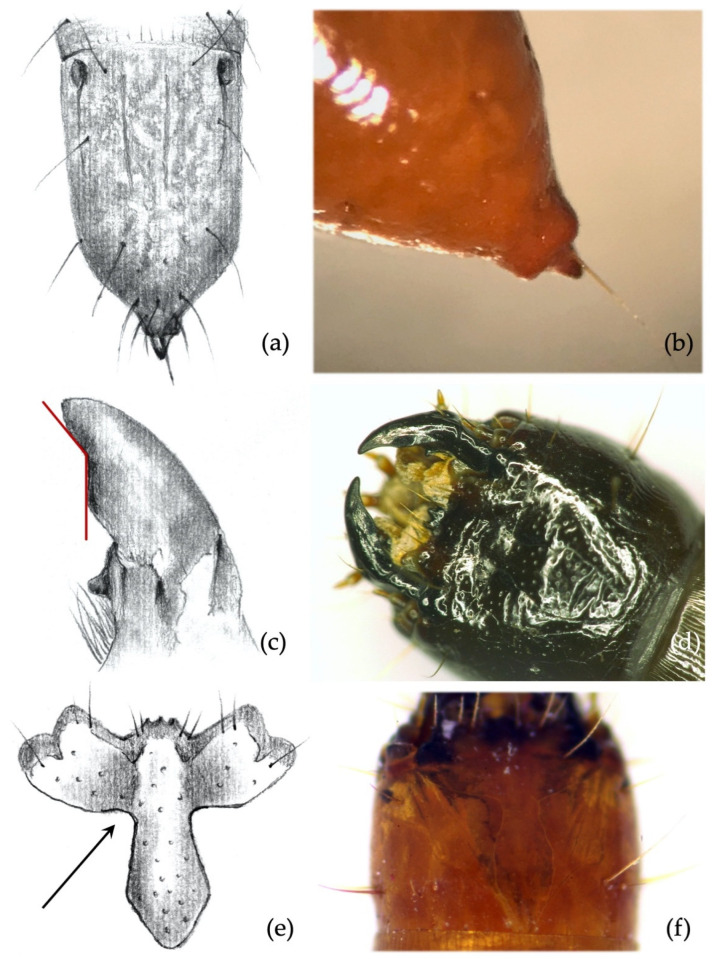
*Agriotes sordidus* (Illiger 1807). Discriminating characters: (**a**,**b**) shape of the caudal part of the 9th abdominal segment with large bulge and terminal spike; (**c**) details of the right mandible with prominent subapical tooth forming an angle far greater than 90°; (**d**) view of the mandibles; (**e**) details of the frontoclypeus, forming a right angle; (**f**) view of the head with visible frontoclypeus.

**Figure 6 insects-12-00534-f006:**
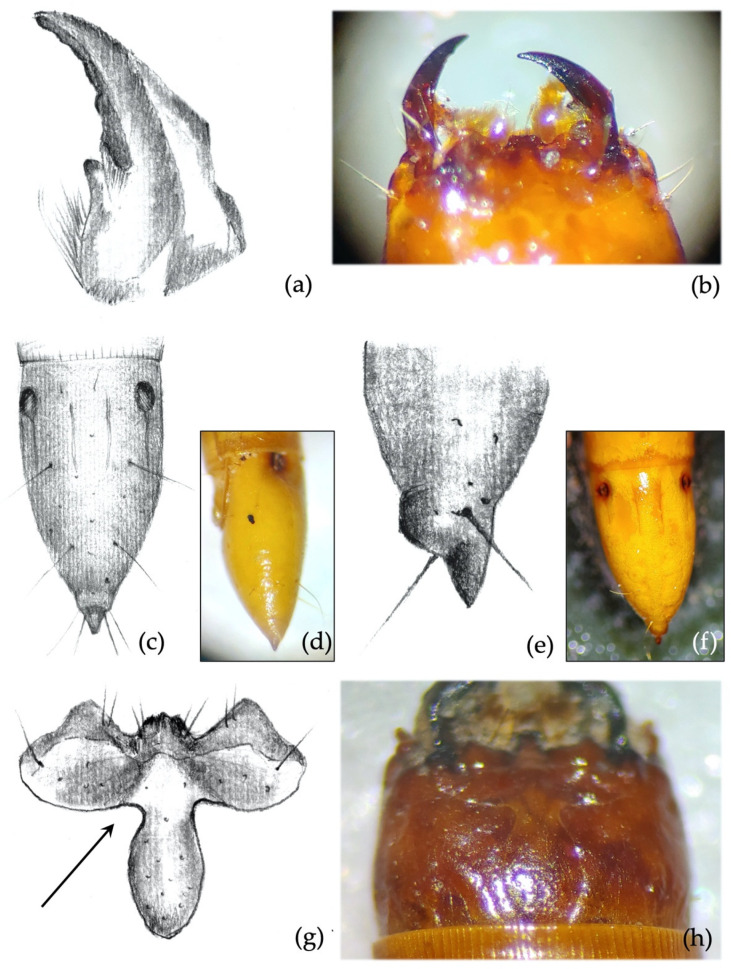
*Agriotes litigiosus* (Rossi 1792). Discriminating characters: (**a**) details of the right mandible; (**b**) view of the mandibles; (**c**) 9th abdominal segment dorsal view; (**d**) 9th abdominal segment side view; (**e**) 9th abdominal segment side view; (**f**) 9th abdominal segment dorsal view; (**g**) details of the frontoclypeus; (**h**) view of the frontoclypeus on the head.

**Figure 7 insects-12-00534-f007:**
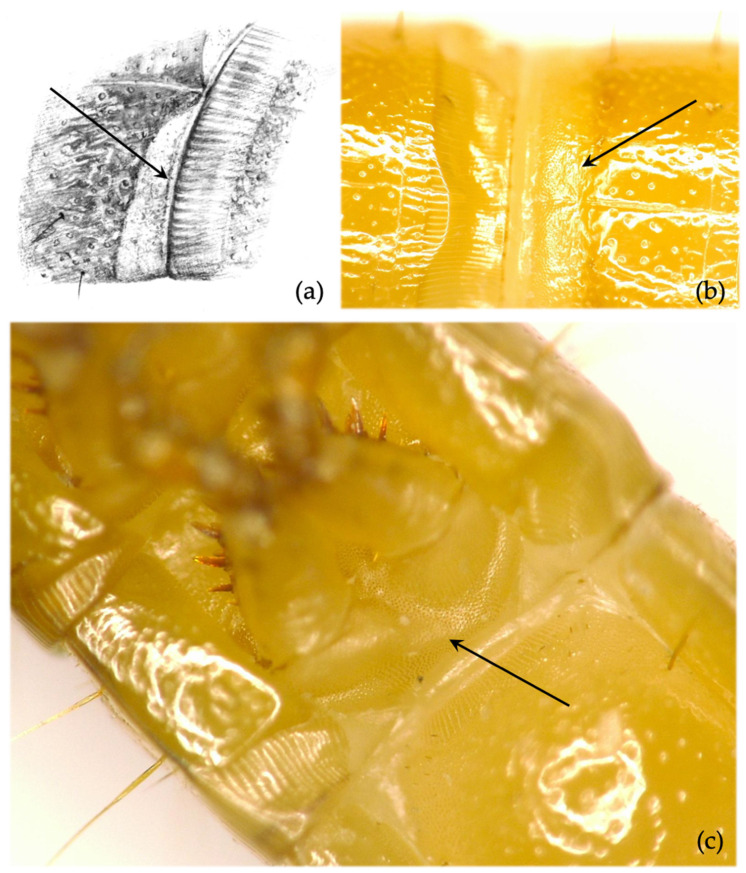
*Agriotes sputator* (L. 1758). Discriminating characters: (**a**,**b**) granules at the anterior margin of abdominal segments (arrows); (**c**) thoracic sternite presenting distinct granules between coxae (arrow).

**Figure 8 insects-12-00534-f008:**
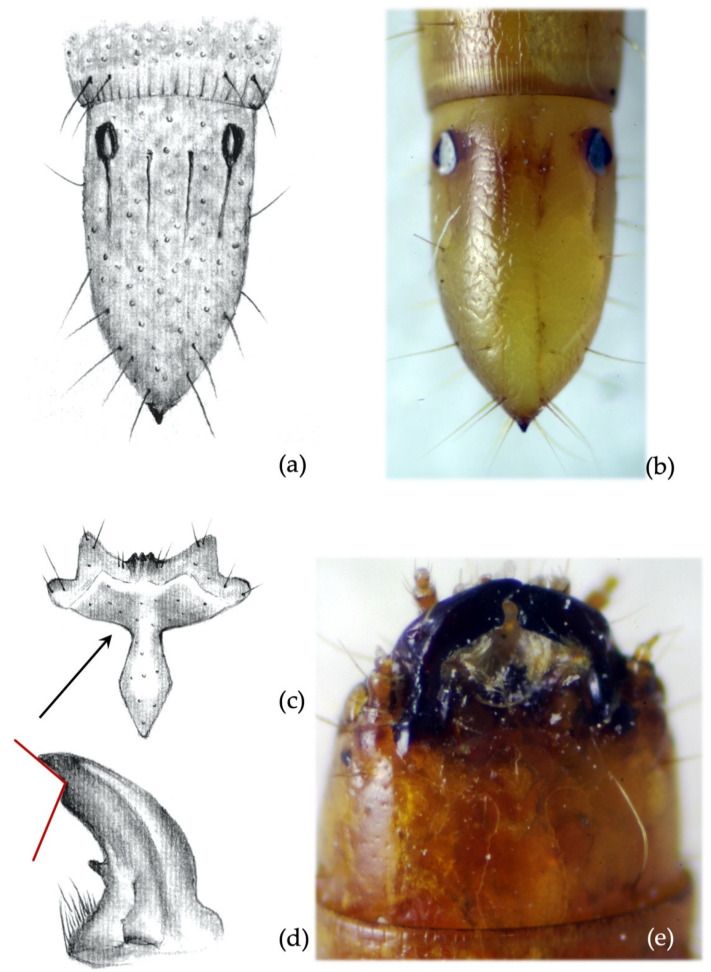
*Agriotes sputator* (L. 1758). (**a**,**b**) Ninth abdominal segment with longitudinal lines and punctures; (**c**) details of the frontoclypeus, forming an angle slightly >90° on at least one side; (**d**) details of the right mandible with subapical tooth forming a measured angle ranging from 90–100°; (**e**) view of the head with frontoclypeus and mandibles.

**Figure 9 insects-12-00534-f009:**
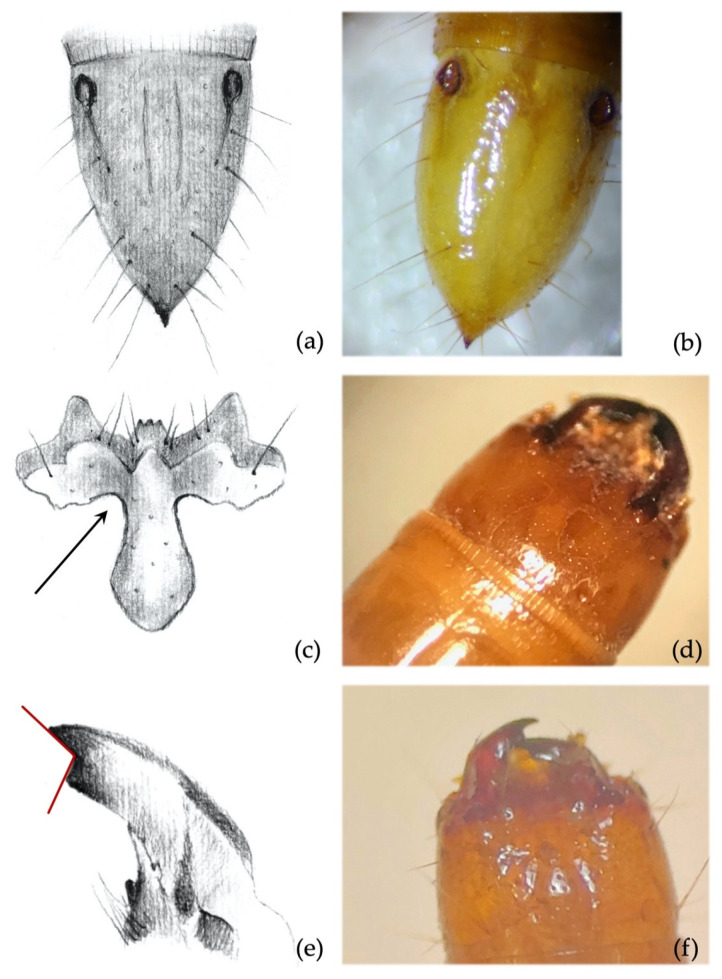
*Agriotes brevis* (Candèze 1863). Discriminating characters: (**a**,**b**) dorsal view of 9th abdominal segment, 1.5× longer than wide; (**c**) details of the frontoclypeus commonly forming an angle <90°; (**d**) view of frontoclypeus on the head; (**e**) details of the right mandible with subapical tooth forming an angle slightly >90°; (**f**) view of mandibles on the head.

**Figure 10 insects-12-00534-f010:**
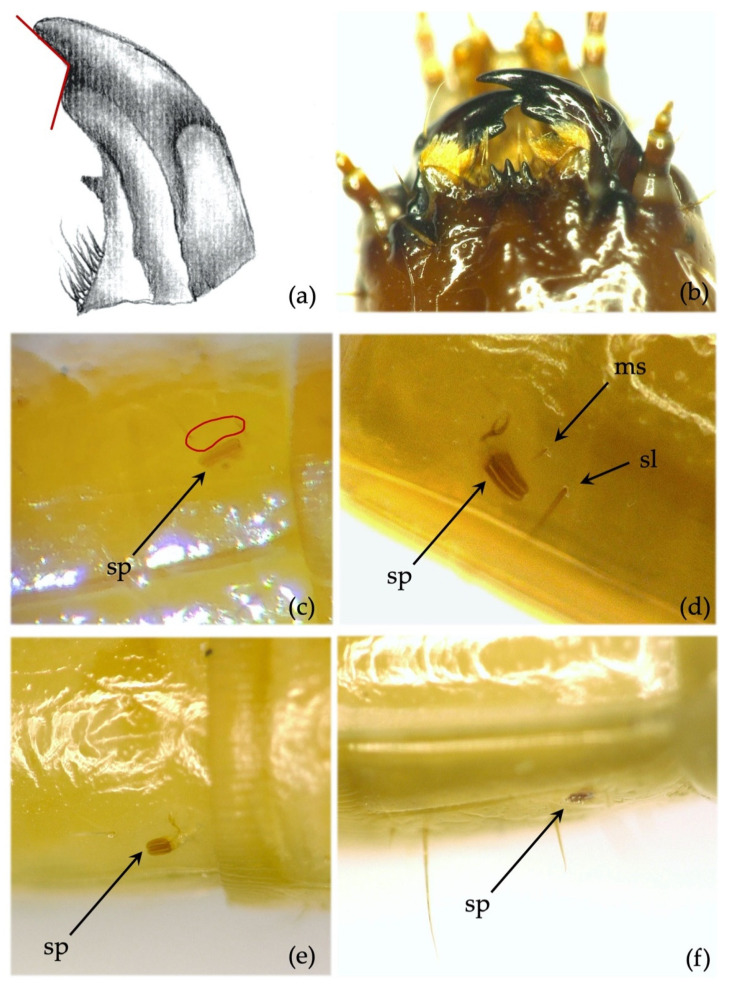
*Agriotes obscurus* (L. 1758) and *Agriotes lineatus* (L. 1767). Discriminating characters: (**a**) details of the right mandible of *A. obscurus* with subapical tooth forming an angle of approximately 90°; (**b**) mandibles of *A. obscurus*; (**c**) lateral view of the abdominal segment of *A. obscurus*: red circle indicates the area where the *minus setae* is located; (**d**) dorso-lateral view of the abdominal segment of *A. obscurus*: ms, *minus setae*; sl, large seta; sp, spiracle; (**e**) lateral view of the abdominal segment of *A. lineatus*; (**f**) dorso-lateral view of the abdominal segment of *A. lineatus*: sp, spiracle.

**Figure 11 insects-12-00534-f011:**
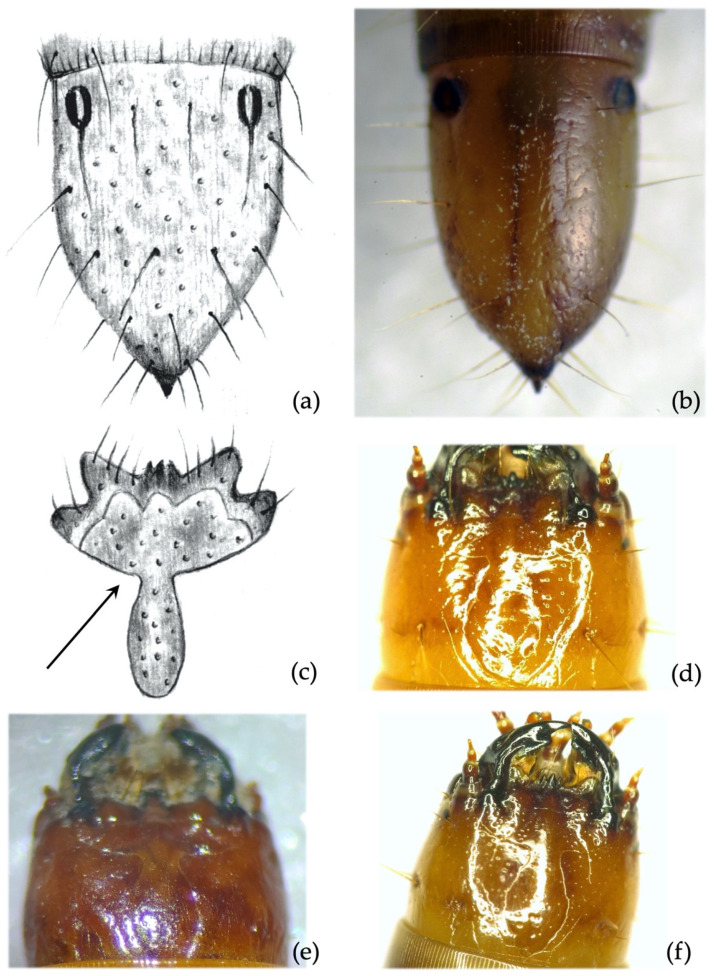
*Agriotes obscurus* (L. 1758). Discriminating characters: (**a**,**b**) 9th abdominal segment, dorsal view; (**c**) details of the frontoclypeus forming an angle from 90–100°. Different aspects of clypeus in different species: (**d**) densely punctured in *A. obscurus*; (**e**) not punctured in *A. litigiosus*; (**f**) scattered punctures in *A. lineatus*.

**Figure 12 insects-12-00534-f012:**
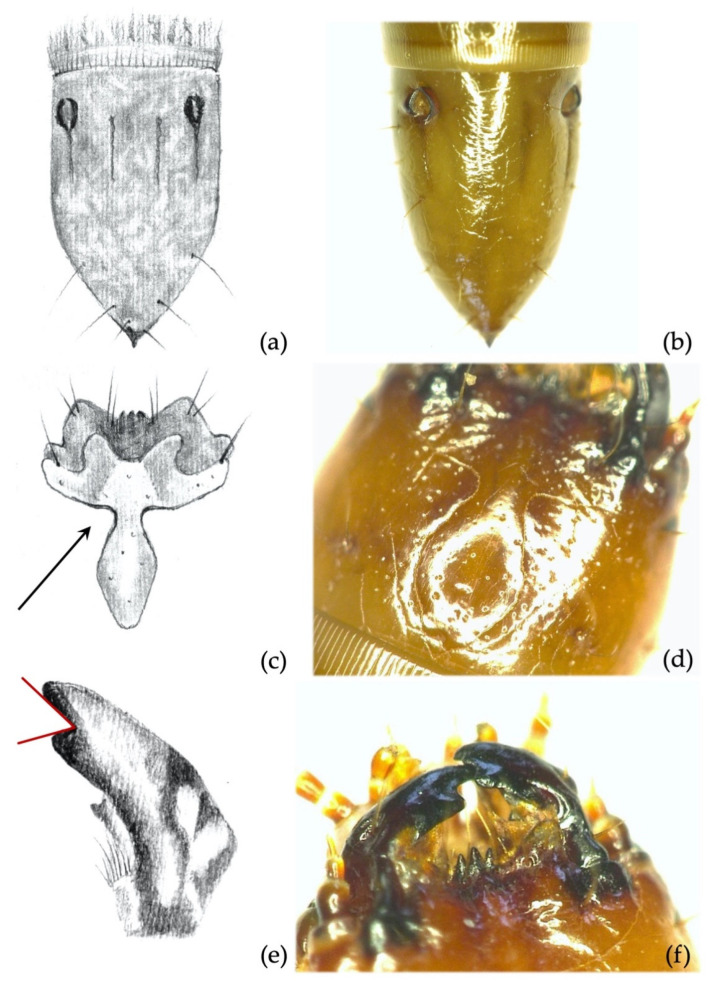
*Agriotes lineatus* (L. 1767). Discriminating characters: (**a**,**b**) 9th abdominal segment dorsal view showing inner and outer longitudinal lines; (**c**,**d**) details of the frontoclypeus forming an angle of about 90°; (**e**) details of the right mandible with subapical tooth, measured angle ≤90°; (**f**) mandibles in dorsal view.

**Figure 13 insects-12-00534-f013:**
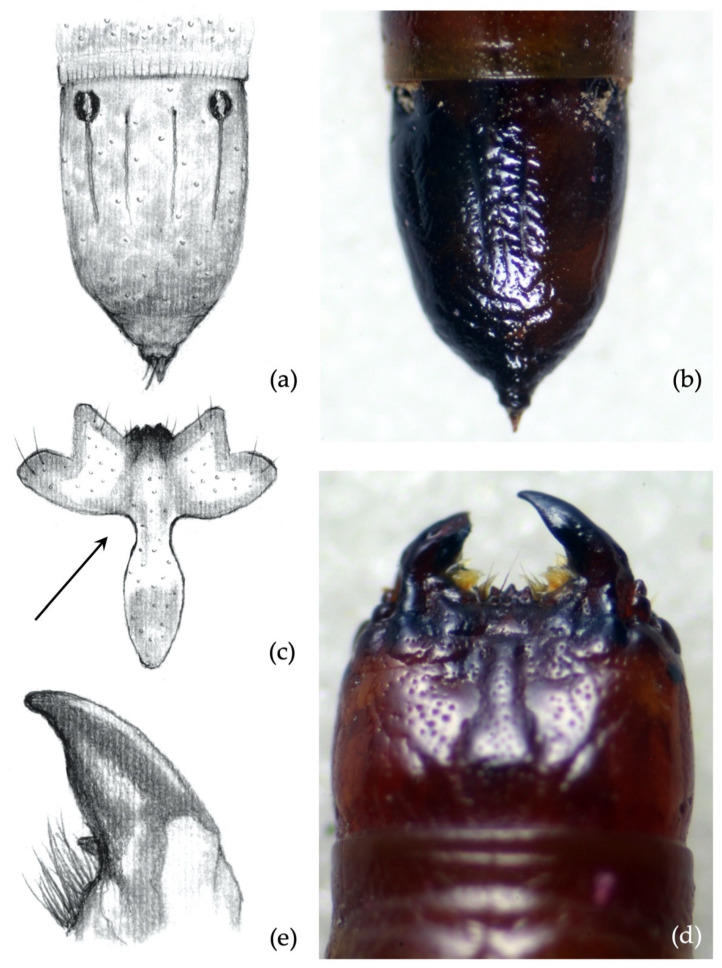
*Agriotes rufipalpis* (Brullé 1832). Discriminating characters: (**a**,**b**) 9th abdominal segment; (**c**) frontoclypeus; (**d**) right mandible; (**e**) head, dorsal view.

**Figure 14 insects-12-00534-f014:**
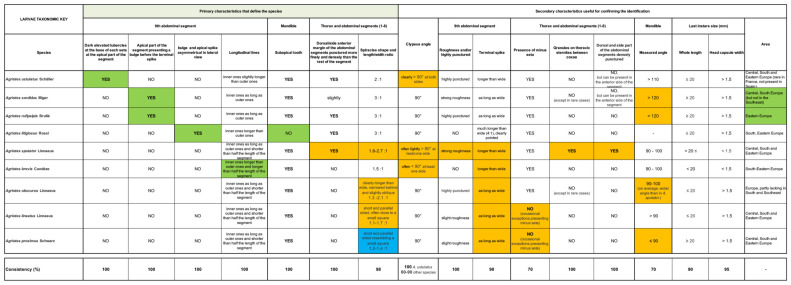
Illustrative image of the horizontal identification table.

## Data Availability

The data presented in this study are available in Appendix A.

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
