# Peer review of "Species Identification of Wireworms (Agriotes spp.; Coleoptera: Elateridae) of Agricultural Importance in Europe: A New “Horizontal Identification Table”"

_insects, 2021, doi:10.3390/insects12060534_

Round 1

Reviewer 1 Report

This paper is well done and allows to identify larvae of several species of Agriotes, considered among the major soil pests of a high number of arable crops, by a key table. Primary and secundary characters are clearly described and illustrated for each species.

However I do not agree with the statement that: this work led to an innovative “horizontal key”. The key used by the authors is simply a modified “modern” synoptic key used in zoology and botany, although much more rarely than a dichotomous key and with another setting suited to a printed journal, since many years, also before the invention of PC.

Currently, the process to report all the known data on a taxon (characters, measures, distribution, bionomics etc.) on Excel is commonly used by many taxonomists and especially by phylogenetists, just for finding apomorphies (unique characters in a group of treated taxa), before a creation of a classic dichotomous key or a phylogenetic tree. It is absolutely right and obviously known by the taxonomists that a lot of information is lost in a dichotomous key, but this type of key can be adapted more easily to a high number of species in a large revision and above all it can be easily reported on a printed journal on the contrary of a “horizonthal key” if too long and complex. Only recently the advent of an online version of the journals has allowed to report such a type of key table although as a online “appendix”.

The final sentence used by the authors “The methodology of this “horizontal key” may be progressively improved by researchers and applied technicians” is right and curiously that which usually concludes many papers where a synoptic key is utilized instead of a dichotomous key.

Therefore I think that the discussion on this type of “synoptic key” (especially the final current sentence) as well as the title and the abstract must be modified accordingly, avoiding the use of the really challenging words “innovative” or “novel”.

Author Response

Reply to reviewer 1

This paper is well done and allows to identify larvae of several species of Agriotes, considered among the major soil pests of a high number of arable crops, by a key table. Primary and secundary characters are clearly described and illustrated for each species.

However I do not agree with the statement that: this work led to an innovative “horizontal key”. The key used by the authors is simply a modified “modern” synoptic key used in zoology and botany, although much more rarely than a dichotomous key and with another setting suited to a printed journal, since many years, also before the invention of PC.

Currently, the process to report all the known data on a taxon (characters, measures, distribution, bionomics etc.) on Excel is commonly used by many taxonomists and especially by phylogenetists, just for finding apomorphies (unique characters in a group of treated taxa), before a creation of a classic dichotomous key or a phylogenetic tree. It is absolutely right and obviously known by the taxonomists that a lot of information is lost in a dichotomous key, but this type of key can be adapted more easily to a high number of species in a large revision and above all it can be easily reported on a printed journal on the contrary of a “horizonthal key” if too long and complex. Only recently the advent of an online version of the journals has allowed to report such a type of key table although as a online “appendix”.

The final sentence used by the authors “The methodology of this “horizontal key” may be progressively improved by researchers and applied technicians” is right and curiously that which usually concludes many papers where a synoptic key is utilized instead of a dichotomous key.

Therefore, I think that the discussion on this type of “synoptic key” (especially the final current sentence) as well as the title and the abstract must be modified accordingly, avoiding the use of the really challenging words “innovative” or “novel”.
R: we thank the reviewer for appreciation; we understand considerations, but based on our research we think that this approach is new for wireworms; we have not found any other taxonomic key with this “horizontal” – multi-discriminating-character approach; we improved the text to make it more clear;

This manuscript represents much new research. This tool would be more useful if the writing were more precise. This key will be useful to many entomologists for many years!
R: provided that we have spent hundreds of hours to make the text logic and concise, we have tried to further improve the manuscript according the reviewer’s suggestions.

Reviewer 2 Report

This manuscript represents much new research. This tool would be more useful if the writing were more precise. This key will be useful to many entomologists for many years!

The writing could be made more concise, better organized, and more informative in many places. Some paragraphs are too long.

Introduction

Please define the scope of applicability of the key. In which European countries can it be used? In which habitats can it be used? Does this apply only to plowed fields/annual crops? Does it work in pastures, hay fields, orchards, vineyards, woodland?

The Introduction needs more specifics establishing that morphological identification of Agriotes larvae to the species level is even possible. The last paragraph suggests this is difficult, and that some authors have made errors. To me it would be useful to mention the success of or problems with other published keys and which characters were problematic.

Why do you call this a ‘horizontal key’? It strongly resembles an ‘identification table’ (e.g. Chrysomelid subfamily key in Stehr, Immature Insects).

Methods

Please state where voucher specimens associating adults and larvae are housed for future studies.

How were illustrations produced (drawings and micrographs).

Results

-Can all larval specimens be identified reliably?

-How do lists of Fundamental characters and secondary characters relate to traditional species descriptions and diagnoses? Why use this format rather than these traditional tools. -What is the minimum set of characters needed to identify each species?

-Would it be useful to add a comments section after each species telling the reader in which regions and habitats the species can be found? You could also add notes about distinguishing that species from similar species.

-Maybe remove the complex numbers before each species.

-Please use only objective descriptive terminology wherever possible.

-Measurements would be more useful if presented as a range of values rather than a single value. For example does ‘2 times’ mean ‘1.95 to 2.05 times’ or ‘1.8 to 2.2 times’?

- Infrasubspecific names are not available. Please do not use them.

-Please always state in which view a structure is observed in both text and figure captions.

-Some head photos are not in focus.

Discussion

Should be edited for clarity and brevity.

Table

-Does ‘X’ mean yes? If so better to say ‘yes’.

-Meaning of orange ‘more discriminating’ is unclear.

-what does yellow mean?

-I don’t understand blue vs. green cells.

Author Response

Reply to reviewer 2 

The writing could be made more concise, better organized, and more informative in many places. Some paragraphs are too long.
R: provided that we have spent hundreds of hours to make the text logic and concise, we have tried to further improve the manuscript according the reviewer’s suggestions;

Introduction
Please define the scope of applicability of the key. In which European countries can it be used?
R: everywhere in Europe apart from Caucaso as stated; we made this more evident;

In which habitats can it be used? 
R: agricultural habitats as stated from the title on;

Does this apply only to plowed fields/annual crops? Does it work in pastures, hay fields, orchards, vineyards, woodland?
R: whatever crop provided they are agricultural crops; when collecting larvae from meadows, rotated pastures and newly ploughed orchard/vineyards grasses, we have always found some of the species included in this key; sometimes more specimens of other genera were found in these crops, but this cannot hamper Agriotes larvae key validity and applicability;  

The Introduction needs more specifics establishing that morphological identification of Agriotes larvae to the species level is even possible. The last paragraph suggests this is difficult, and that some authors have made errors. To me it would be useful to mention the success of or problems with other published keys and which characters were problematic.
R: this is already discussed; we have improved as much as possible;

Why do you call this a ‘horizontal key’? It strongly resembles an ‘identification table’ (e.g. Chrysomelid subfamily key in Stehr, Immature Insects).
R: thanks for the suggestion, we modified it in “horizontal identification table”;

Methods
Please state where voucher specimens associating adults and larvae are housed for future studies.
R: DONE

How were illustrations produced (drawings and micrographs).
R: we have introduced specific authors; 

Results
-Can all larval specimens be identified reliably?
R: yes

-How do lists of Fundamental characters and secondary characters relate to traditional species descriptions and diagnoses? Why use this format rather than these traditional tools. -What is the minimum set of characters needed to identify each species?
R: we give for any species several discriminating characters with their consistency; this makes possible to get a reliable identification; the best table use is described alongside the minimum set of character to achieve a reliable species identification;

-Would it be useful to add a comments section after each species telling the reader in which regions and habitats the species can be found? 
R: the key is intended to be applied to any agricultural field wherever in Europe; we do not have scientifically sound information to discriminate between habitats as declared from the title on, as we have focused on Agriotes species of agricultural importance;

You could also add notes about distinguishing that species from similar species.
R: we present a complete, detailed table for this; 

-Maybe remove the complex numbers before each species.
R: DONE 

-Please use only objective descriptive terminology wherever possible.
R: DONE 

-Measurements would be more useful if presented as a range of values rather than a single value. For example does ‘2 times’ mean ‘1.95 to 2.05 times’ or ‘1.8 to 2.2 times’?
R: DONE

- Infrasubspecific names are not available. Please do not use them.
R: DONE; we have deleted the reference to var. laichartingi; this does not change anything from a practical point of view. The name was used without any taxonomic meaning but instead as a simple categorization to split the two forms.

-Please always state in which view a structure is observed in both text and figure captions.
R: DONE 

-Some head photos are not in focus.
R: replaced 

Discussion
Should be edited for clarity and brevity.
R: the text has been improved.

Table
-Does ‘X’ mean yes? If so better to say ‘yes’.
R: DONE 

-Meaning of orange ‘more discriminating’ is unclear.
R: we have simplified color use and added a caption 

-what does yellow mean?
R: we simplified color use and added a caption 

-I don’t understand blue vs. green cells.
R: we simplified color use and added a caption

Round 2

Reviewer 1 Report

I agree with the modification of the text according to my suggestions

Reviewer 2 Report

Changes implemented, thanks.

Suggestions
-'minus setae' to 'minor setae'. Here major vs. minor.
-no 'roundish' tubercle. This is the default state, and round could have several meanings depending on viewing direction.
-characters could be 'primary' vs. 'secondary'. 'Fundamental' sounds almost religious.
-'spine' is standard for entomology, plants have 'thorns', 'spike' is unnecessary dupication.
-'alive larvae' to 'live larvae'
-' straightaway' to 'immediately.